# Infantile Anemia and Iron Treatments Affect the Gut Microbiome of Young Rhesus Monkeys

**DOI:** 10.3390/microorganisms13102256

**Published:** 2025-09-26

**Authors:** Christopher L. Coe, Gabriele R. Lubach, Wellington Z. Amaral, Gregory J. Phillips, Mark Lyte, Michael K. Georgieff, Raghavendra B. Rao, James R. Connor

**Affiliations:** 1Harlow Center for Biological Psychology, University of Wisconsin, Madison, WI 53715, USA; grlubach@wisc.edu; 2California Department of Health Care Access and Information, Sacramento, CA 95833, USA; wellingtonamaral@gmail.com; 3Department of Infectious Diseases, University of Georgia, Athens, GA 30602, USA; greg.phillips@uga.edu; 4Department of Veterinary Microbiology and Preventative Medicine, Iowa State University, Ames, IA 50011, USA; mlyte@iastate.edu; 5Department of Pediatrics, University of Minnesota, Minneapolis, MN 55455, USA; georg001@umn.edu (M.K.G.); raghurao@umn.edu (R.B.R.); 6Department of Neurosurgery, Penn State University College of Medicine, Hershey, PA 17033, USA; jconnor@pennstatehealth.psu.edu

**Keywords:** microbiome, anemia, iron, ferritin, infancy, rhesus monkey

## Abstract

The influence of iron deficiency anemia and iron treatments on the gut microbiome was evaluated in young rhesus monkeys. First, the hindgut bacterial profiles of 12 iron-deficient anemic infants were compared to those of 9 iron-sufficient infants at 6 months of age, a time when the risk of anemia is high due to rapid growth. After this screening, the anemic monkeys were treated with either parenteral or enteral iron. Seven monkeys were injected intramuscularly with iron dextran, the typical weekly treatment used in veterinary practice. Four other anemic infants were treated with a novel oral supplement daily: yeast genetically modified to express ferritin. Fecal specimens were analyzed using 16S ribosomal RNA (rRNA) gene amplicon sequencing. Bacterial species richness in anemic infants was not different from that of iron-sufficient infants, but beta diversity and LEfSe analyses of bacterial composition indicated that the microbiota profiles were associated with iron status. Both systemic and oral iron increased alpha and beta diversity metrics. The relative abundance of Ruminococcaceae and other Firmicutes shifted in the direction of an iron-sufficient host, but many different bacteria, including Mollicutes, Tenericutes, and Archaea, were also enriched. Collectively, the findings affirm the important influence of the host’s iron status on commensal bacteria in the gut and concur with clinical concerns about the possibility of adverse consequences after iron supplementation in low-resource settings where children may be carriers of iron-responsive bacterial pathogens.

## 1. Introduction

Iron is an essential micronutrient for all prokaryotic and eukaryotic life, serving as a critical cofactor that catalyzes enzymatic reactions, oxidative metabolism, and protein synthesis and is needed to support immune defense [1,2,3]. Despite iron’s abundance in the environment, however, not all forms of iron are soluble and bioavailable, and when bacteria colonize larger host organisms, they become reliant on the more proximal sources of iron in the host’s diet and heme iron [4,5]. The clinical significance of the complex interactions between the host’s iron and commensal bacteria becomes more evident in the context of iron deficiency anemia, as well as after iron treatments and diseases that increase iron levels in the digestive tract [6]. It is estimated that over 80% of oral iron supplements remain unabsorbed by the host and can become available to siderophilic bacteria in the large intestine [7]. It is less clear how parenterally administered iron influences the intestinal microbiome, and if so, whether these changes are like the effects of enteral supplementation. Many clinical studies have reported that iron deficiency is associated with an increased risk of diarrheal disease in undernourished individuals [8,9]. In addition, some iron treatments used to remedy anemia can have the unwanted consequence of worsening a pre-existing gut dysbiosis and inflammatory bowel conditions [10,11,12].

The dynamics of iron acquisition and utilization in the host can also be influenced by the gut microbiota [4,13]. Because iron is an essential element, bacteria make use of multiple high-affinity ferric iron chelators, such as enterobactin, to competitively acquire and solubilize iron for use when concentrations are low [14,15]. Conversely, in the large intestine, where iron availability may be higher, bacteria can tightly regulate iron uptake to avoid toxicity from its redox potential [16]. The mechanisms that bacteria employ to scavenge iron from the host in both low- and high-iron conditions via siderophore chelation, as well as directly from heme iron via hemophores, have been extensively interrogated using both in vitro and ex vivo colonic models [17,18,19,20]. In general, most mechanistic studies of iron acquisition have focused on bacterial pathogens, because siderophore systems are prominent in Enterobacteriaceae, including *E. coli*, *Shigella* spp., *Yersinia* spp., and *Klebsiella* spp. [21,22]. Beyond the long-standing concerns about iron supplementation in parts of the world where malaria is endemic, the importance of iron usage by many enteric bacterial pathogens, including *Campylobacteria jejuni*, has been well characterized and includes promoting growth and pathogenicity and enabling a competitive advantage over non-siderophilic taxa in the commensal community.

Significant effects of infantile anemia and iron treatments on the gut microbiome have also been found using in vivo models, although the magnitude and even the direction of the changes in specific taxa vary across rodent and porcine studies [23,24,25,26,27,28]. These findings have important practical applications for animal husbandry with domesticated piglets, because, like humans, they are especially prone to iron deficiency anemia due to their rapid growth and the low bioavailability of iron in the mother’s milk [29]. Our research extends the comparative perspective to monkeys who also birth large babies predisposed to a postnatal infantile anemia if the gravid female consumed a non-fortified diet while pregnant [30,31]. We employed a standardized experimental protocol that results in 20–30% of infant rhesus monkeys becoming iron-deficient anemic (IDA) by 6 months of age when their mothers consume a diet with only moderate iron levels (225 mg Fe/kg of diet) [32]. Their fetuses receive less maternal iron prenatally, and a subset are at risk for IDA as they deplete iron reserves to support rapid growth, exceeding the bioavailable sources of iron in breast milk [33]. This maternal-fetal-neonatal iron dynamic mimics the situation in humans around the world, where rates of maternal iron deficiency during pregnancy can approach 60%, even in well-resourced countries [34].

In addition to comparing the microbial profiles of iron-deficient infants to iron-sufficient monkeys, we examined the effects of two iron treatments. Some anemic monkeys were treated by intramuscular injection of iron dextran, the typical weekly regimen used in veterinary medicine [35,36]. Iron dextran has been used clinically for anemic patients since the 1950s and was the only parenteral iron preparation approved for human use in the United States until 2000 [37,38,39]. Other formulations, such as iron sucrose, ferric gluconate, and ferric carboxymaltose, have been approved more recently for intravenous administration. In 1996, the intramuscular administration of iron dextran was approved by the Food and Drug Administration (FDA) for the routine prevention and treatment of iron-deficiency anemia in domesticated farm animals [40]. In addition, we conducted an exploratory evaluation of a novel oral supplement: yeast genetically modified to express ferritin [41]. Because ferritin is the intracellular and extracellular protein used by the body to sequester and safely store iron in a non-toxic form in tissue sites and macrophages, it may potentially lessen its bioavailability to enteropathogens [42]. However, some pathogenic bacteria, including *Salmonella enterica* and enterotoxigenic *Escherichia coli*, can utilize ferritin- and transferrin-bound iron [43,44,45]. In addition, other opportunistic pathogens like *Pseudomonas aeruginosa* can synthesize ferritin and bacterioferritin directly [46,47]. Given the known links between host iron and gut bacteria, we investigated whether raising iron levels in systemic circulation by injection or providing ferritin in the digestive tract would differentially affect the hindgut microbiota profiles of young monkeys.

## 2. Materials and Methods

### 2.1. Specimen Collection, Animal Husbandry, and Source

Blood and fecal specimens were collected from 21 young rhesus monkeys (14F, 7M, *Macaca mulatta*). These infants were born in an established breeding colony housed in a large indoor facility with standardized husbandry conditions [48]. They were full-term and raised naturally by multiparous females, who lived socially as pairs while pregnant and nursing their young. All adult monkeys were fed the same diet, which was manufactured specifically for primates with a targeted amount of iron (225 mg Fe/kg, Lab Diet 5LFD, St Louis, MO, USA). This diet provides sufficient iron for a non-pregnant monkey but is not additionally fortified with the extra iron required to entirely fulfill maternal/fetal needs before term [33]. It was the predisposing factor that created this experimental model of infantile IDA, which does not require any postnatal manipulations. Each adult female was given approximately 250 g of biscuits daily. Additional details about the biscuit composition and diet are provided in Appendix A. The light/dark schedule was kept constant at 16 h light/8 h dark, with lights on at 0600, a photoperiod that overrode the inherent tendency of rhesus monkeys to breed seasonally [49]. It ensured that pregnancies and births could be scheduled year-round and that sample collections from the growing infants were synchronized. Both blood and fecal samples were obtained in the morning between 0900 and 1100. The husbandry and sample collection protocols were approved by the institutional Animal Care and Use Committee (ACUC) at the University of Wisconsin-Madison.

### 2.2. Experimental Design

The iron status of the infant monkeys was determined from small (3 mL) heparinized blood samples collected via femoral venipuncture when they were 6 months of age. Complete blood counts, including hemoglobin (HgB) and mean corpuscular volume (MCV) tests, were ordered at a clinical laboratory familiar with nonhuman primate specimens (Unity Meriter Lab, Madison, WI, USA). The criteria for diagnosing iron-deficiency anemia (IDA) in these infant rhesus monkeys included: HgB < 100.0 mg/L and MCV < 60.0 fL. In contrast, normal hematological values for 6-month-old, iron-sufficient infants (IS) were: HgB > 114.0 mg/dL and MCV > 69.5 fL.

After the initial baseline assessment, the anemic infants were assigned to two different iron treatment conditions. Seven were administered weekly intramuscular injections of iron dextran (10 mg, Phoenix Pharmaceutical, St. Joseph, MO, USA) until their hematological indices returned to the typical range, which occurred within 1–2 months. In keeping with veterinary practice, the iron dextran regimen also included a B vitamin complex (see Appendix A for details). Iron dextran is a colloidal solution of ferric oxyhydroxide complexed with low-molecular-weight, polymerized dextran. The form of iron is ferric (Fe^3+^), and the composition allows for a slow release of iron, minimizing the toxicity that can arise from free iron. When administered intramuscularly, iron dextran bypasses the gastrointestinal tract, is largely bound to transferrin in the circulation, and is transported via the reticuloendothelial system, where it is transported to the liver, spleen, and bone, and recycled via macrophages. The dextran component of the complex ensures that iron is released slowly, reducing the risk of sudden increases in free iron that could lead to cellular damage. Significant increases in hemoglobin levels are generally seen within two to four weeks.

The second treatment condition involved an oral iron supplement: nutritional yeast (*Saccharomyces cerevisiae*) genetically modified to express high levels of ferritin. Yeast transformed with H-ferritin was shown to be well-tolerated by mice, rats, and monkeys, and an evaluation of yeast with ferritin that had been labeled with a stable iron isotope verified that the ^57^Fe was absorbed from the gut and incorporated into red blood cells [41]. Four infants were given a daily dose of 6 mg ferritin/kg body weight based on the typical dosage prescribed for ferrous sulfate when given orally to treat iron-deficient monkeys. Voluntary consumption of the powdered yeast was facilitated in a non-stressful manner by mixing it with a small amount of jam or peanut butter. Infants evincing the lowest hematological values, indicative of more severe anemia (i.e., MCV < 50 fL), were assigned to the iron dextran condition, taking their welfare into consideration. Thus, while all iron-deficient infants met the criteria for anemia prior to treatment, the hematological parameters of the 4 receiving oral supplements were initially not as low. During the treatment period, 1–2 fecal samples were obtained from each monkey with sterile cotton swabs and rapidly frozen in an ultracold freezer at <70 °C.

### 2.3. Yeast Construction and Preparation

Construction and preparation of the yeast-ferritin complex (YFC) followed established methods and have been in detail previously [41]. Briefly, a 600 bp human Fth1 fragment was cloned into plasmid pL5652 and integrated at the yeast TDH3 promoter in strain BY4741 to produce strain P3190. To characterize the human heavy chain ferritin (Fth1) expressed in the P3190 yeast cell line, it was grown in high iron-containing media, and after cell lysis, the Fth1 protein was purified from soluble lysates to >90%. Based on ferrozine assays, it contained significant amounts of iron, estimated at ~110 Fe atoms per subunit. Size Exclusion Chromatography column-purified fractions analyzed by electron microscopy revealed 4–8 nm dense particles with the properties expected for Fth1 as a 24-mer. Collectively, the analysis indicated that the Fth1 in this yeast formed native oligomers containing non-mineralized iron-rich cores.

### 2.4. DNA Extraction from Fecal Specimens and Library Preparation

DNA was extracted from the fecal specimens at the Wright Lab (Huntingdon, PA, USA) using a DNeasy Powersoil Kit (Qiagen, Germantown, MD, USA) according to the manufacturer’s protocol and eluted in 50 μL of DNase/RNase-free water. Purified DNA was quantified using an Invitrogen Qubit 4 Fluorometer and 1× dsDNA High Sensitivity Assay Kit (ThermoFisher Scientific, Waltham, MA, USA). PCR primers 515f/806r were used to amplify the variable region 4 of the 16S rRNA gene. 16S rRNA libraries were created using Illumina-tag PCR reactions with the DNA extracts following the Earth Microbiome Project’s protocol [50]. PCR products were pooled and gel-purified on a 2% agarose gel using the Qiagen Gel Extraction Kit (Qiagen, Germantown, MD, USA). Before sequencing, the purified pool was quality-checked using an Agilent 2100 BioAnalyzer and Agilent DNA High Sensitivity DNA kit (Agilent Technologies, Santa Clara, CA, USA). The purified pool was initially stored at −20 °C and then sequenced using an Illumina MiSeq Reagent Kit v2 with paired-end 250 base pair reads.

#### 16S rRNA Amplicon Sequencing

Raw data were imported into QIIME2 for processing and analysis [51]. Initial quality in the form of Phred q scores was determined using QIIME2, while the cumulative expected error for each position was determined with VSEARCH [52]. Primers (515F/806R) were removed from all reads. Then, forward reads were truncated to a length of 249, and reverse reads were truncated to a length of 225, with a maximum expected error of 0.5 for both, as implemented in QIIME2’s DADA2 pipeline [53]. QIIME2’s DADA2 pipeline was also used to merge forward and reverse reads, remove chimeras, and assign the remaining sequences to amplicon sequence variants (ASVs). Representative sequences were used for taxonomic assignments of the ASVs, using a Naive Bayes classifier as trained in QIIME2’s “qiime feature-classifier classify-sklearn” on the Silva v132 OTUs database trimmed to the 515F/806R region. Representative sequences were also employed to create a rooted phylogenetic tree using MAFFT and FastTree through QIIME2’s “qiime phylogeny align-to-tree-mafft-fasttree” [54,55].

Alpha diversity was calculated to capture the richness, evenness, observed features, and phylogenetic metrics of taxa present in each sample [56], and beta diversity metrics were used to examine the similarity or dissimilarity of bacterial communities between experimental conditions. Alpha diversity was calculated by subsampling the ASV table at 10 different depths, ranging from 2300 to 23,000 sequences, using QIIME2’s diversity alpha rarefaction. Twenty iterations were performed at each depth to generate an average alpha diversity value. A rarefaction plot was created from the results of this subsampling to confirm that diversity approached an asymptote and that the slope decreased as depth increased. The richness metric (Chao 1) was representative and selected for presentation. Beta diversity was assessed using a Cumulative Sum Scaling (CSS) normalization to account for differences in sequencing depth [57,58]. Weighted Unique Fraction (UniFrac) distances were calculated from the normalized table and rooted phylogenetic tree [59,60]. The resulting distance matrix was visualized as a Principal Coordinates Analysis (PCoA) plot in QIIME2’s Emperor.

Linear discriminant analysis Effect Size (LEfSe) was used to identify taxa with abundances that differed between IDA and IS infants, as well as after iron treatment of the anemic monkeys [61]. To explore whether differences in hindgut bacterial profiles might influence the gut milieu or host physiology, Kyoto Encyclopedia of Genes and Genomes (KEGG) functional pathways were inferred using the Phylogenetic Investigation of Communities by Reconstruction of Unobserved States (PICRUSt) tool from the bioBakery suite [62,63,64,65].

### 2.5. Significance Testing

Differences in iron-related hematological values between IDA and IS infants were compared with *t*-tests, and the improvement in iron status after the two iron treatments was examined using a two-way, repeated measures analysis of variance (ANOVA) followed by post hoc testing. Significance of differences in alpha diversity between conditions was examined with pairwise *t*-tests. while differences in beta diversity were assessed with ANOSIM. Taxonomic representation between conditions was assessed using LEfSe. Taxa were considered enriched if log(LDA) Score ≥ 2.0 and Kruskal–Wallis *p* ≤ 0.05 [66].

## 3. Results

### 3.1. Hematology of Iron Sufficient (IS) and Iron Deficient Anemic (IDA) Infants at Baseline and Post-Treatment

Screening 6-month-old rhesus monkeys for hematological criteria of anemia yielded two distinct subgroups of IDA and IS infants with non-overlapping blood values. Anemic monkeys had significantly lower HgB levels and smaller MCVs than the IS monkeys (Table 1). The body weights of the IDA monkeys did not differ significantly when compared to the IS infants (1.351 ± 0.216 g vs. 1.418 ± 0.172 g, respectively), nor did the anemic monkeys show overt physical signs of poor health or diarrheic symptoms. Following treatment with intramuscularly injected iron dextran or oral supplementation with the YFC, there were significant increases in HgB levels (Table 1). Improvements in MCV values were more gradual and attained statistical significance only in the anemic monkeys administered iron dextran, not in the ones consuming the yeast-ferritin supplement. The more rapid increase in the size of red blood cells after injection of iron dextran was expected, even though all 7 of these anemic infants initially had lower MCV values prior to treatment (between 41.3 and 49.4 fL).

### 3.2. Microbiota Profiles of IS and IDA Infants

Rectal swabs were obtained from the IS and IDA infants on the same day that blood was collected, and the DNA extracted from fecal specimens was assessed with 16S bacterial rRNA gene amplicon sequencing. Sequencing data from all 21 monkeys was included in the analyses of taxonomic abundance and calculation of diversity metrics. Alpha diversity did not differ based on the iron status of the IS and IDA monkeys prior to iron treatment. The comparable richness was due in part to the extent of variation in the number of taxa identified across monkeys (Figure 1A). However, the analysis of beta diversity did still indicate that the microbial profiles of IS and IDA monkeys were dissimilar, reflecting a divergence in community structure (*p* = 0.0509) (Figure 1B). *Faecalibacterium*, butyrate-producing Gram-positive anaerobes in the Family Ruminococcaceae (Class Clostridia), were among the taxa found to be present in higher abundance in IS infants (Figure 1C). In contrast, among the IDA infants, *Veillonella* and *Streptococcus*, two potentially pathogenic genera in the Phylum Bacillota, were enriched, along with a higher abundance of Gammaproteobacteria, a Gram-negative class within the Phylum Pseudomonadota.

### 3.3. Microbial Responses to Iron Treatment of IDA Infants

Weekly injections with iron dextran resulted in an increased number of bacterial species identified in the hindgut of anemic monkeys, which accounted for a significant difference in alpha diversity (*p* < 0.02) (Figure 2A). The community structure also appeared to be more convergent during the treatment phase. ANOSIM testing of the dissimilarity between pre- and post-treatment, based on the presence and abundance of taxa, indicated there was a significant difference in beta diversity (*p* < 0.002) (Figure 2B). When the relative abundances of specific taxa were interrogated more closely, iron treatment was found to be associated with an enrichment of Ruminococcaceae, a bacterial family that had been more abundant in IS infants. However, additional taxa were also uniquely enriched by the increase in systemic iron levels, including Tenericutes, Mollicutes, and Archaea (Figure 2C). At the phylum level, Firmicutes were more prevalent during the treatment phase, accompanied by a relative reduction in Bacteroidetes, which had been higher prior to treatment. The latter shift included a displacement in the relative abundance of Prevotellaceae (Appendix A). During the period of iron treatment, the higher relative abundances of Methanomassiliicoccaaceae and Thermoplasmata were especially notable.

Oral iron supplementation also increased bacterial species richness in the hindgut, resulting in a significant difference in alpha diversity after consumption of yeast expressing ferritin (YFC) (Figure 3A). Although the sample size is small, repeat specimens were obtained from each monkey during the supplementation period. The beta diversity analysis indicated that the community structure was also different post-treatment, and the gut microbiota profiles of the supplemented monkeys appeared more convergent (Figure 3B). LEfSe analyses indicated several similarities with the response to injected iron dextran, including an enrichment of *Ruminococcus*, Mollicutes, and Tenericutes (Figure 3C). Additional phylogenetic differences at the phylum level are shown in cladogram format in Appendix A and include an enrichment of Clostridia within the Phylum Firmicutes. There were also some notable differences in the microbial profile after YFC supplementation when directly compared with the post-treatment gut microbiome after iron dextran (see Appendix A). After injection of iron dextran, the enrichment of the methane-producing Methanomassiloiococcaceae and Archaea was more pronounced. Within the Archaea, the methane-producing and sulfur-reducing Class Thermoplasmata was also more prominently enriched by administering iron dextran. Further, an enhancement of Euryachaeorta and Porphyromonadacae was evident. Conversely, the post-treatment comparison of microbial profiles indicated that the relative abundance of Erysipelotrichaceae remained more salient in anemic monkeys supplemented orally with YFC (Appendix A). Despite these differences, however, the post-treatment community structures were similar overall, and there was no significant difference in beta diversity when the treatment effects of iron dextran were directly compared with those of yeast–ferritin (ANOSIM, *t* = 0.062, *p* = 0.193 NS).

### 3.4. Predictions of Functional Differences Based on Bacterial Gene Profiles

In addition to the enrichment of methane-producing Archaea, which included the sulfur-reducing Class Thermoplasmata in the iron dextran condition, the directed microbial genomic changes could result in several metabolic and physiological effects. First, PICRUSt was employed to infer pathways that might already be different in anemic monkeys before treatment when compared to IS monkeys. Many bacterial genes associated with cellular processes, information processing, metabolism, and host–pathogen interactions were differentially expressed (Figure 4). These predictive analyses suggested that the gut microbiome of IS infants would be more supportive of pathways associated with endocrine and immune functioning in the host, whereas aspects of the bacterial profiles of the IDA infant would be more likely to activate primary and secondary pathways associated with bile acid biosynthesis. Pathways involved in glycan biosynthesis and metabolism were already lower in IDA than IS infants and appeared to be further downregulated after administration of iron dextran to anemic monkeys (Appendix A). In contrast, following oral consumption of YFC, there was a more pronounced enhancement of microbial genes favoring pathways associated with fructose, mannose, galactose, starch, and sucrose metabolism when compared to the gut microbiota profiles of anemic monkeys before iron supplementation (Appendix A). However, these inferences derive from changes in bacterial composition and abundance, and only some of the predicted metabolic effects have been directly assessed with physiological assessments (e.g., [67]).

## 4. Discussion

Our evaluation of the gut microbiota of anemic monkeys concurs with prior research showing that iron deficiency and iron treatments can affect the hindgut microbiome [68,69,70]. The findings also highlight the importance of the host’s diet and heme iron as key sources of this essential micronutrient for commensal symbionts [71]. However, it should be acknowledged that the magnitude and extent of the effect of iron-limited and iron-rich conditions on specific bacterial taxa vary across animal models of anemia, with different taxa impacted in rodent and porcine studies [72,73]. In addition, the unwanted consequences of enteral iron treatments on the gut microbiome may also be distinct in undernourished children living in low-resource settings, where they are more likely to be infected with enteric pathogens [74,75]. Many pathogenic bacteria, including *E. coli*, *S. enterica*, and *S. flexneri*, have effective chelating mechanisms, including siderophores and hemophores, that facilitate the acquisition of iron from the host, which enable them to proliferate with increased pathogenicity in iron-rich conditions [76,77].

One strength of our study was that it was conducted in a hygienic laboratory setting. While several bacterial families enriched in anemic monkeys include pathogenic species, such as Streptococcacaeae, and some of the enriched genera like *Veillonella* can be associated with inflammatory conditions, diarrheic symptoms were not observed prior to or after administration of iron. Further, it should be reiterated that our experimental paradigm targeted a single micronutrient deficiency and was not a model of undernutrition or growth stunting. The body weights of the anemic infants were comparable to the iron-sufficient monkeys. In fact, both the IDA and IS monkeys were substantially heavier by 300–400 g than similarly aged rhesus monkeys after their growth had been impeded by symptomatic infections with *C. jejuni*, *S. flexneri*, enterotoxigenic *E. coli,* or *Giardia* in a more naturalistic outdoor setting [78].

One bacterial genus that prominently distinguished the gut microbiome of anemic from iron-sufficient monkeys was the reduction in the relative abundance of *Faecalibacterium*, a member of the Family Ruminococcaceae. While this family is not a predominant one in rhesus monkeys, with a relative abundance that is only between 3 and 5% in infant rhesus monkeys of this age [79,80,81], it is a major producer of butyrate and other short-chain fatty acids (SCFAs). In human patients, lower levels of *Faecalibacterium* have been associated with inflammatory gut conditions, including IBD [82,83,84]. In addition, another member of the same Class Clostridia, *Clostridiacheae SMB53*, was also less abundant in IDA than in iron-sufficient infants. It is a consumer of gut mucus and plant-derived saccharides and was previously found to be enhanced after copper supplementation in growing piglets [85]. Conversely, in the IDA infants, Gammaproteobacteria were relatively more abundant. This class includes many Gram-negative pathogens, such as *E. coli* and *S. flexneri*, both of which can become a cause of symptomatic diarrhea in monkeys. Pre-treatment differences in the gut microbiota of anemic monkeys and children may increase the risk of pathogenic Gram-negative enteric bacteria emerging in an iron-rich setting [74]. For example, we found that the relative abundance of Erysipelotrichaceae was higher in the anemic monkeys prior to treatment with YFC, and then the difference became more salient during the oral supplementation period. In humans, Erysipelotrichaceae are a taxon found to be opportunistically associated with inflammatory gut conditions [86].

Many bacteria can tolerate low-iron conditions. For example, Proteobacteria can use secreted hemophores to directly extract iron from the host’s transferrin, ferritin and lactoferrin [1,87]. Prior to treatment, *Haemophilus* and *Actinobaccilus*, two genera within Pasteurellaceae, were relatively more abundant in the IDA monkeys. In addition, although not typically considered pathogenic in monkeys, the genus *Veillonella* (Phylum Bacillota) was consistently more abundant in the anemic infants. These bacteria were also more abundant in one study of anemic children [75], although not consistently found to be different in all investigations of the gut microbiome in iron-deficient children [88]. *Veillonella* are also known to be responsive to high lactate conditions. In rodent models of anemia, it was shown that iron-deficient rats have higher blood lactate levels than iron-replete animals, especially when engaging in aerobic activity [89]. An influence of high lactate on *Veilllonella* was the explanation used to account for their relatively higher abundance in fecal specimens collected from marathon athletes [90]. In addition, previous metabolomic analyses of anemic monkeys have revealed a bioenergetic imbalance within the central nervous system, suggestive of suppressed tricarboxylic acid (TCA) cycle activity and increased glycolysis [91]. Anemic monkeys had a lower citrate/lactate ratio in their cerebrospinal fluid, indicative of increased cerebral glucose uptake and lactate production.

A tolerance of some bacteria for low-iron conditions may be relevant for understanding the inconsistent findings on whether anemia and iron treatments affect two beneficial commensal bacteria, *Bifidobacteria* and *Lactobacilli* [92,93]. We identified both genera in the hindgut of the infant monkeys, but their abundances were relatively low at the end of the nursing period and were not overtly associated with iron status. Differences in findings on *Bifidobacteria* and *Lactobacilli* across studies could reflect the use of 16S rRNA gene amplicon sequencing and a reliance on calculating relative abundances if less prevalent bacteria become displaced by the enrichment of other taxa during iron treatments. *Bifidobacteria* are tolerant of low-iron conditions, and *Lactobacilli* are known to be more dependent on magnesium than iron [94]. Therefore, neither may be especially vulnerable to iron deficiency prior to iron treatments and during iron-rich conditions both may be prone to being displaced by other taxa.

After administration of iron dextran and oral YFC supplementation, there was a more salient enrichment of Ruminococcaceae, which had initially been found to be present at a relatively lower abundance in anemic than iron-sufficient infants. The post-treatment shift in *Ruminococcus* contributed to an overall increase in the relative abundance of Firmicutes at the phylum level. There were also several unique changes, including the enrichment of Archaea, Tenericutes, and Mollicutes. The increase in anaerobic methanogenic Archaea warrants further investigation, because they are known to be the primary producers of methane in the digestive tracts of ruminant animals [95,96] and can have both positive and negative effects on human health [97,98].

The mechanistic pathways through which parenterally and enterally administered iron affects bacteria in the large intestine still need to be more systematically delineated. In humans, excess iron is not actively excreted via the large intestine, and the body’s iron pool is regulated primarily by controlling absorption at the level of the small intestine and by reticuloendothelial recycling [99,100]. However, in mice, rats, and to some degree in dogs, it has been found that an increase in systemic iron is reflected by increased iron concentrations in feces, which indicates there may be species differences in iron clearance and excretion [101,102,103]. But the more likely pathway is that the administered iron raised hepcidin levels, the primary regulator of iron absorption in the small intestine, allowing more dietary iron to remain in the gut and reach the colonic bacteria [104,105,106]. Alternatively, the influence of parenteral iron on gut bacteria could be due to increased levels of free iron in the blood and higher intraluminal iron concentrations. Further, during the treatment period, there would have been more iron-containing hemoglobin and a higher oxygen-carrying capacity of the RBC in the mesenteric arteries and vasculature villi of the large intestine. The changes in bacterial composition and species richness observed after the provision of enteral ferritin also need further investigation. While intracellular storage of ferritin in macrophages and sequestration of iron as ferritin in tissue can limit the bioavailability of iron for bacteria in circulation or after a breakdown in the barrier protection of the skin, when ferritin was provided directly via oral supplementation, it appeared to enrich many of the same taxa as parenteral iron. In human volunteers, it was also shown that when oral iron was given as ferrous sulfate, it increased fecal concentrations of iron, which then elevated the production of free radicals in the colon [107].

Although these findings in anemic monkeys are novel, several limitations of the research should be acknowledged. The number of anemic infants was small for considering two treatment conditions, even though the total sample size is typical for an experiment with monkeys. In addition, our model focused on a single micronutrient deficiency, whereas it is more common for anemia in low-resource settings to be associated with a more pervasive undernutrition. But one important feature of our paradigm is that it did not require any postnatal manipulations or restriction of the nursing infant’s diet. The risk for infantile anemia was created by limiting the iron content of the mother’s diet during pregnancy.

Our experimental design was also cross-sectional, and the assessment age was selected to coincide with when infant monkeys are most likely to be anemic. However, some microbial differences may be antecedent and precede the onset of anemia. Research with other animal models of anemia has shown that bacteria higher up in the small intestine can also influence the bioavailability of iron, in part through their effects on gut acidity, which then facilitates or lessens iron absorption [108,109,110]. In the current study, we also did not track long-term outcomes beyond the treatment period. We already knew from prior research that the hematological anemia in infant monkeys would completely resolve over time with the consumption of solid foods [32], even though some metabolic effects persist [67,91]. For example, it is of interest that the PICRUSt analyses suggested there might be an effect on primary and secondary bile acid pathways. A prior metabolomic analysis of blood from anemic infant monkeys documented higher serum concentrations of 7α-hydroxy-3-oxo-4-cholestenoic acid (7-HOCA) and chenodeoxycholate, and lower concentrations of three different bile acids, taurocholate, taurodeoxycholate, and tauroursodeoxycholate relative to age-matched, iron-replete monkeys. The possibility of microbial effects on bile acid pathways is in keeping with a growing awareness of the important connections between gut bacteria, bile synthesis, and lipid metabolism [111,112]. Our previous metabolomic analyses also revealed that 323 other metabolites continued to be differentially expressed after the remediation of anemia with iron dextran, including ones associated with monoamine synthesis and metabolism [113]. The latter difference may be attributable to persistent alterations in the synthesis of key amino acid precursors by gut bacteria. Lastly, we should acknowledge the limitations of 16S bacterial gene amplicon sequencing for making functional predictions based on relative abundance and community structure. Other methods, such as shotgun metagenomics and metatranscriptomics, can provide more precise species-level identification and enable more accurate predictions of gene function pathways [64,114,115,116].

## 5. Conclusions

Notwithstanding these caveats, which can be addressed in future research, our findings demonstrate that monkeys are a useful animal model for investigating infantile anemia. For example, prior to treatment, anemic monkeys were found to have a lower relative abundance of *Faecalibacterium* and other Ruminococcaceae, important producers of SCFA. In addition, the enrichment of methanogenic Archaea, Tenericutes, and Mollicutes that was elicited by two different modes of iron treatment warrants further investigation. There is now a need to examine the other iron formulations used for intravenous administration in clinical practice to manage anemia that is unresponsive to oral treatment in children and adults.

## Figures and Tables

**Figure 1 microorganisms-13-02256-f001:**
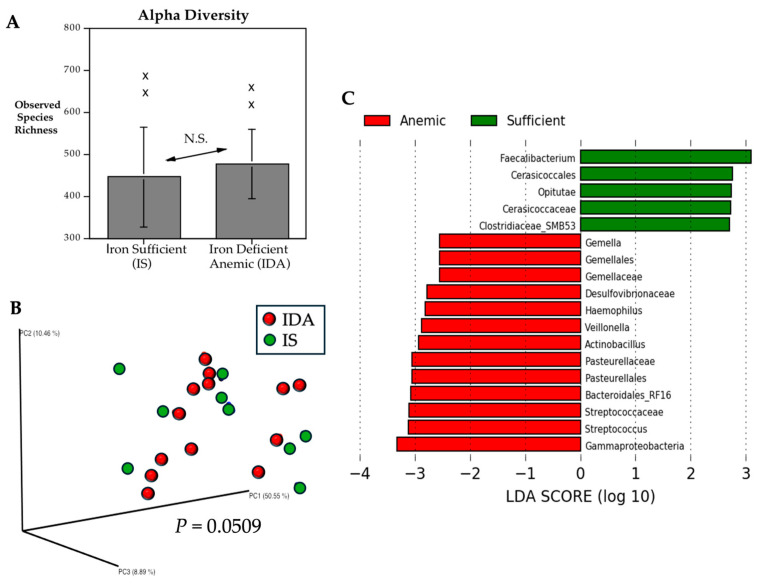
Gut microbiome of anemic monkeys (IDA) compared to iron-sufficient (IS) infants at 6 months of age. There was considerable individual variation in bacterial richness and, as conveyed by the arrows and N.S., alpha diversity (Chao 1) did not differ significantly between conditions. Two divergent higher values in each condition are shown with X symbols (**A**). However, the bacterial composition of the hindgut of IDA monkeys (red circles) did differ from the microbial profiles of IS infants (green circles), resulting in a significant difference in beta diversity (**B**). The distance matrix is visualized as a PCoA plot. LEfSe analyses indicated that the relative abundance of many taxa differed between IDA and IS monkeys (**C**). Only log LDA differences exceeding 2.0 are shown; dotted lines show the units for the LDA scores. Taxonomic differences are also portrayed in a cladogram format in Appendix A, illustrating other distinctive phylogenetic features.

**Figure 2 microorganisms-13-02256-f002:**
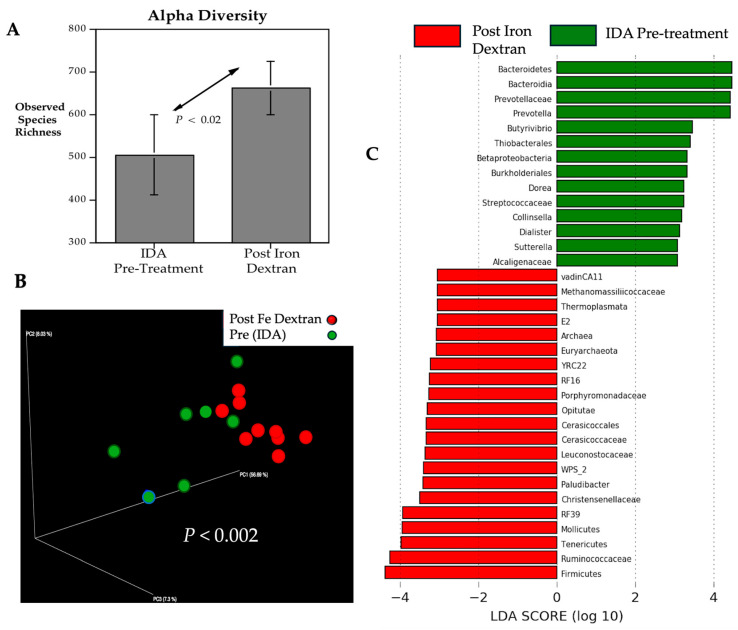
Microbial profiles of anemic monkeys at baseline and after administration of iron dextran. Iron treatment resulted in a significant increase (*p* < 0.02) in alpha diversity (Chao 1) (**A**). The PCoA plot illustrates the clustering of similar taxa and the increased concordance in abundance post-treatment, resulting in a significant difference (*p* < 0.002) in beta diversity (**B**). Many taxa were enriched post-treatment, attaining significant log(LDA) scores > 2.0, *p* < 0.05 (**C**). The phylogenetic shift at the phylum level post-treatment included a relative increase in the abundance of Firmicutes, whereas prior to treatment, there was a relatively higher abundance of Bacteroidetes (see Appendix A).

**Figure 3 microorganisms-13-02256-f003:**
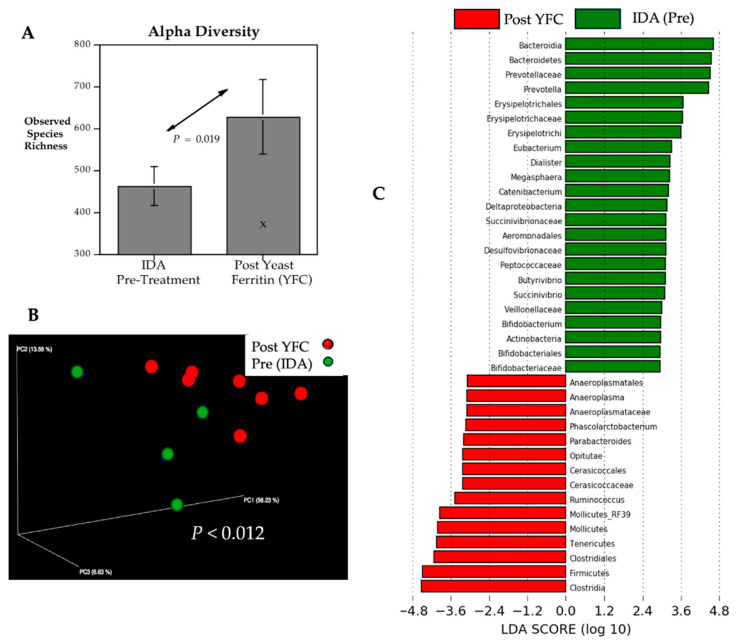
Microbial profiles of anemic monkeys prior to treatment and after oral supplementation with yeast expressing ferritin. During the supplementation phase, two fecal samples were obtained from each monkey; however, sequencing from one specimen was excluded due to a low number of ASVs, resulting in 7 determinations. Despite the small N, there appeared to be a significant increase (*p* < 0.019) in alpha diversity (Chao 1) after treatment (**A**). One sample with a low species richness post-treatment value is indicated with an X symbol. There also appeared to be more concordance in microbial community structure and a significant difference in beta diversity (**B**). Pretreatment and posttreatment samples are shown with green and red circles, respectively. Overall, the shifts in alpha and beta diversity followed a pattern like those observed after iron dextran administration. In addition, the LEfSe analysis indicated the relative abundance of several taxa changed similarly, including an enrichment of *Ruminococcus*, Mollicutes, and Tenericutes (**C**). A cladogram illustration of the taxonomic changes at the phylum level is presented in Appendix A, showing the relatively increased abundance of Firmicutes and Tenericutes, whereas Bacteroidetes and Actinobacteria were more prevalent in anemic monkeys prior to the treatment.

**Figure 4 microorganisms-13-02256-f004:**
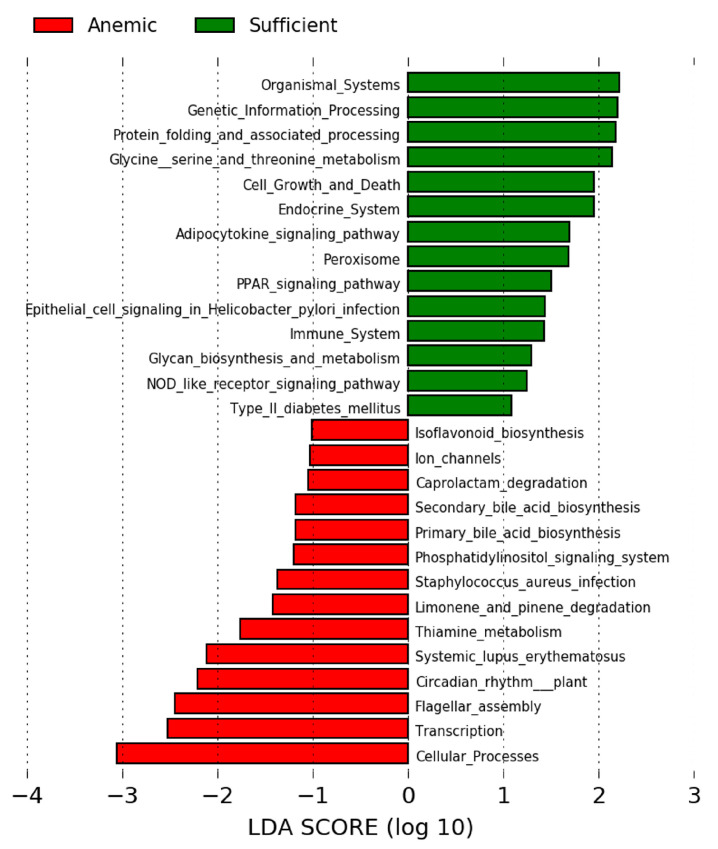
Functional pathways were inferred using PICRUSt to examine potential metabolic differences between anemic and iron-sufficient infants. Predictions were based on taxa with LDA scores that exceeded 2.0 (shown in Figure 1C). The gut microbiome of iron-sufficient infant monkeys may exert a stronger influence on their endocrine and immune physiology. Conversely, in anemic infants, there were more microbial genes that could potentially affect primary and secondary bile acid biosynthesis. In keeping with this prediction, prior metabolomic evaluations of anemic monkeys have identified differences in the circulating levels of bile acids as a prominent serum biomarker associated with anemia [67].

**Table 1 microorganisms-13-02256-t001:** Hematology of Iron Sufficient (IS) and Iron-Deficient Anemic (IDA) monkeys, before and after treatment with iron dextran (Fe Dextran) or consuming yeast expressing ferritin (YFC).

Hematology	IS	IDA	*p* *	Post-Fe Dextran	*p* **	Post-YFC	*p* ***
HgB (mg/L)	127.1	85.1	0.001	132.5	0.001	117.3	0.03
(SD)	(9.1)	(8.9)		(10.1)		(9.3)	
MCV (fL)	70.8	46.5	0.001	63.7	0.001	57.45	NS
(SD)	(1.2)	(4.7)		(5.8)		(1.3)	

* IDA compared to IS monkeys. ** Pre (IDA) vs. post after weekly intramuscular injections of anemic monkeys with iron dextran. *** Pre (IDA) vs. post after daily oral supplementation of anemic monkeys with Yeast–Ferritin Complex.

## Data Availability

The original contributions presented in this study are included in the article/Appendix A. Further inquiries can be directed to the corresponding author.

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
