# Peer review of "Infantile Anemia and Iron Treatments Affect the Gut Microbiome of Young Rhesus Monkeys"

_microorganisms, 2025, doi:10.3390/microorganisms13102256_

Round 1

Reviewer 1 Report

Comments and Suggestions for Authors

The manuscript titled with “Infantile anemia and iron treatments affect the gut microbiome of young rhesus monkeys” systematically investigated the influence of iron deficiency anemia and iron treatments on the gut microbiome in young rhesus monkeys. This is an interesting study that seems to be novel and well conducted. However, there are some points in the current version to be needed correction. This manuscript could be accepted after minor revision.

Specific comments:

  1. The description of the experimental design in the abstract was not clear enough. For instance, among the 21 experimental animals, only the treatment methods of 11 of them were described. What were the treatment methods for the remaining animals? Furthermore, the description of the test results is not detailed enough.
  2. L107: This title does not match the content. It needs to be revised.
  3. L154:“>70℃”?Is it right?
  4. The conclusion section did not provide a comprehensive summary of the research results, and references should not be included in the conclusion.

Author Response

Reviewer 1

We appreciate that Reviewer 1 felt that the study was novel and conducted well. We have tried to be responsive to each of the concerns and requests. Below we detail the specific changes.

  1. The description of the experimental design in the abstract was not clear enough. For instance, among the 21 experimental animals, only the treatment methods of 11 of them were described. What were the treatment methods for the remaining animals? Furthermore, the description of the test results is not detailed enough.

We have modified the text in the Abstract to present the experimental plan more clearly.  We more clearly state that only the anemic infants were administered iron.  This aspect of the experimental design was in keeping with the approval by the institutional Animal Care and Use Committee.  Based on that approval and animal welfare considerations, we treated all infant monkeys found to be anemic with iron until their hematology values showed that they now in the normal range. With respect to reporting more findings in the Abstract, we are limited by the journal’s strict Word Limits.  At this point, we cannot add more information this section without exceeding the journal guidelines.

Abstract

The influence of iron deficiency anemia and iron treatments on the gut microbiome was evaluated in young rhesus monkeys. First, the hindgut bacterial profiles of 12 iron-deficient anemic infants were compared to 9 iron-sufficient infants at 6 months of age, when the risk for anemia is high due to rapid growth. After this screening, the anemic monkeys were treated with either parenteral or enteral iron. Seven monkeys were injected intramuscularly with iron dextran, the typical weekly treatment used in veterinary practice. Four other anemic infants were treated with a novel oral supplement daily: yeast genetically modified to express ferritin. Fecal specimens were analyzed using 16S ribosomal RNA (rRNA) gene amplicon sequencing. Bacterial species richness in anemic infants was not different from iron-sufficient infants, but beta diversity and LEfSe analyses of bacterial composition indicated that the microbiota profiles were associated with iron status. Both systemic and oral iron increased alpha and beta diversity metrics. The relative abundance of Ruminococcaceae and other Firmicutes shifted in the direction of an iron-sufficient host, but many different bacteria, including Mollicutes, Tenericutes, and Archaea, were also enriched. Collectively, the findings affirm the important influence of the host’s iron status on commensal bacteria in the gut and concur with clinical concerns about the possibility of adverse consequences after iron supplementation in low resource settings where children may be carriers of iron-responsive bacterial pathogens. 

  1. L107: This title does not match the content. It needs to be revised.

As requested, the heading for this subsection has been modified.  It is now:  Specimen collection, animal husbandry and source

2. L154:“>70℃”?Is it right?

Thank for noticing the typing error.  It has been corrected.  It now reads < 70 oC because all samples were stored in an ultracold freezer below this temperature.

3. The conclusion section did not provide a comprehensive summary of the research results, and references should not be included in the conclusion.

The references and discursive sentences have been removed from the Conclusion section.  We added another sentence describing the findings but also endeavored to keep this section succinct because the findings had been reported in Results and just been discussed.

Line 527-535  Revised Conclusions section.

Notwithstanding these caveats, which can be addressed in future research, our findings demonstrate that monkeys can provide a useful animal model for investigating infantile anemia. For example, prior to treatment, anemic monkeys were found to have a lower relative abundance of Faecalibacterium and other Ruminococcaceae, important producers of SCFA. In addition, the enrichment of methanogenic Archaea, Tenericutes and Mollicutes that was elicited by two different modes of iron treatment warrants further investigation. There is now a need to examine the effect of intravenous administration because it has become the recommended clinical method for the management of anemia unresponsive to oral treatment in children and adults

As mentioned in the Response letter to the journal, the figures have been reformatted and are now presented in a clearer manner with a larger font size in the graphic illustration of the LEfSe analyses.  The column headings and footnotes for Table 1 now more explicitly convey more that the hematology values post-treatment refer just to the anemic infants after they have been treated.

Reviewer 2 Report

Comments and Suggestions for Authors

The authors' findings indicate that monkeys serve as a valuable animal model for studying infantile anemia. Notably, the increase in methanogenic Archaea, Tenericutes, and Mollicutes observed with two different modes of iron treatment calls for further investigation. The authors also emphasize the need to explore the effects of intravenous administration, as this is the recommended clinical approach for managing anemia that does not respond to oral treatment in both children and adults.

Furthermore, the potential impact of microbial activity on bile acid pathways aligns with the growing understanding of the critical connections between gut bacteria, bile synthesis, and lipid metabolism. Additionally, the effects of iron deficiency on the activity of monoamine neurotransmitters may be partly attributed to anemia's influence on the bacterial synthesis of amino acid precursors for serotonin and dopamine.

The authors highlight the significant impact of the host's iron status on the composition of commensal bacteria in the gut. They also share concerns regarding the possible negative consequences of iron supplementation, especially in low-resource settings, where children may be carriers of iron-responsive bacterial pathogens.

Strength of the manuscript: the authors proved that monkeys can be adequate models for the study of infantile anemia.

Weakness of the manuscript: the authors could have given some assumed model schematically where they would indicate different ways of iron application.

Recommended corrections:

  1. In the introductory part, something should be written about siderophilic bacteria.
  2. In the introductory part, it is necessary to indicate: In most aerobic environments, such as the soil or sea, iron exists in the ferric (Fe3+) state, which tends to form insoluble rust-like solids. To be effective, nutrients must not only be available, but they must also be soluble. Microbes release siderophores to scavenge iron from these mineral phases by forming soluble Fe3+ complexes that active transport mechanisms can take up.
  3. Give an example of the structure of a bacterial siderophore where Fe3+ is chelated.
  4. In the introductory part, write more about how Fe3+ is complexed in humans, and then write a little more about ferritin.
  5. To explain how iron dextran was selected, it is necessary to characterize: what form the iron is in, if it is in iron oxide, then explain how it turns into a dissolved Fe3+ ion, and the dextran polymer also needs to be characterized.
  6. Give composition and characterization: nutritional yeast (Saccharomyces cerevisiae).
  7. Can bile salts with alpha OH groups and with a carboxylate group from the side chain complex the Fe3+ ion so that the Fe3+ ion would participate in the enterohepatic circulation.
  8. The figures from the manuscript would be attractive, but especially the C and D parts are not visible, the markings and text are unreadable (Fig 1-3).

Author Response

We appreciate that Reviewer 2 felt that we had provided convincing evidence that the monkey provides a valuable model for investigating infantile anemia and iron treatments.  Most importantly, we have reformatted the figures to that they have only 3 panels, rather than 4 illustrations.  That allowed us to increase the size of the panel showing the results of the LEfSe analysis and permitted a more legible font size.  We have  provided additional information that the reviewer requested.

  1. In the introductory part, something should be written about siderophilic bacteria.

The introductory text on siderophilic bacteria has been lengthened.  There is now a full paragraph (lines 70-85).

The dynamics of iron acquisition and utilization in the host can also be influenced by the gut microbiota [4, 13]. Because iron is an essential element, bacteria make use of multiple high affinity ferric iron chelators, such as enterobactin, to competitively acquire and solubilize iron for use when concentrations are low [14,15]. Conversely, in the large intestine, where iron availability may be higher, bacteria can tightly regulate iron uptake to avoid toxicity from its redox potential [16]. The mechanisms that bacteria employ to scavenge iron from the host in both low and high iron conditions via siderophore chelation ,as well as directly from heme iron via hemophores, have been extensively interrogated using both in vitro and ex vivo colonic models [17-20]. In general, most mechanistic studies of iron acquisition have focused on bacterial pathogens, because siderophore systems are prominent in Enterobacteriaceae, including E. coli, Shigella spp, Yersinia spp, and Klebsiella spp. [21,22]. Beyond the long-standing concerns about iron supplementation in parts of the world where the Plasmodium malarial parasite is endemic, the importance of iron usage by many enteric bacterial pathogens, including Campylobacteria jejuni, has been well characterized and includes promoting growth and pathogenicity and enabling a competitive advantage over non-siderophilic taxa in the commensal community.

2. In the introductory part, it is necessary to indicate: In most aerobic environments, such as the soil or sea, iron exists in the ferric (Fe3+) state, which tends to form insoluble rust-like solids. To be effective, nutrients must not only be available, but they must also be soluble. Microbes release siderophores to scavenge iron from these mineral phases by forming soluble Fe3+ complexes that active transport mechanisms can take up.

We agree that is an important point.  We have changed the second sentence to now state ‘soluble and bioavailable’. Lines 56-57.

3. Give an example of the structure of a bacterial siderophore where Fe3+ is chelated.

We provide ‘enterobactin’ as an example of a bacterial siderophore.

4. In the introductory part, write more about how Fe3+ is complexed in humans, and then write a little more about ferritin.

Additional information has been provided in both the Introduction. and Methods section.  As noted, below in response to item 6, we also now provide information on the construction of the yeast-ferritin construct.

Lines 110-118

In addition, we conducted an exploratory evaluation of a novel oral supplement: yeast genetically modified to express ferritin [41]. Because ferritin is the intracellular and extracellular protein used by the body to sequester and safely store iron in a non-toxic form in tissue sites and macrophages, it could potentially lessen its bioavailability to enteropathogens [42]. However, some pathogenic bacteria including Salmonella enterica and enterotoxigenic Escherichia coli can utilize ferritin- and transferrin-bound iron [43-45]. In addition, other opportunistic pathogens like Pseudomonas aeruginosa can synthesize ferritin and bacterioferritin directly [46,47].

5. To explain how iron dextran was selected, it is necessary to characterize: what form the iron is in, if it is in iron oxide, then explain how it turns into a dissolved Fe3+ ion, and the dextran polymer also needs to be characterized.

Additional information is provided in both the Introduction and Methods.

Lines 103-110 Introduction

Some anemic monkeys were treated by intramuscular injection of iron dextran; the typical weekly regimen used in veterinary medicine [35,36]. Iron dextran has been used clinically for anemic patients since the 1950s and was the only parenteral iron preparation approved for human use in the United States until 2000 [37-39]. Other formulations, such as iron sucrose, ferric gluconate, and ferric carboxymaltose have been approved more recently for intravenous administration. In 1996 intramuscular administration of iron dextran was approved for the routine prevention and treatment of clinical anemia in domesticated farm animals [40].

Lines 152-165 Methods

After the initial baseline assessment, the anemic infants were assigned to two different iron treatment conditions. Seven were administered weekly intramuscular injections of iron dextran (10 mg, Phoenix Pharmaceutical, St. Joseph, MO, USA) until their hematological indices returned to the typical range, which occurred within 1-2 months. In keeping with veterinary practice, the iron dextran regimen also included a B vitamin complex (see Table S2 for details). Iron dextran is a colloidal solution of ferric oxyhydroxide complexed with low molecular weight, polymerized dextran. The form of iron is ferric (Fe3+), and the composition allows for a slow release of iron, minimizing the toxicity that can arise from free iron. When administered intramuscularly, iron dextran bypasses the gastrointestinal tract, is largely bound to transferrin in circulation and passaged via the reticuloendothelial system, where it is transported to the liver, spleen and bone, and recycled via macrophages. The dextran component of the complex ensures that iron is released slowly, reducing the risk of sudden increases in free iron that could result in cellular damage. Significant increases in hemoglobin levels are generally noted within two-to-four weeks.

If iron is released from macrophages, it is released via ferroportin as Fe+2 and converted to Fe+3 by hephastin bound to ferroportin and subsequently bound primarily to plasma proteins like transferrin in circulation.  We share the reviewer’s interest in this aspect of iron metabolism and regulation, but given the current paper’s focus on the gut microbiome, we felt a need to keep the text focused on those endpoints.

6. Give composition and characterization: nutritional yeast (Saccharomyces cerevisiae).

We had previously referred the reader to an earlier publication that described the construction and preparation of the yeast-ferritin complex.  However, we appreciate why it is also of value to present a succinct version within the current report.  The following section and new text are now in the Methods section.

2.3. Yeast construction and preparation

YFC construction and preparation followed established methods and have been reported in detail previously [41]. Briefly, a 600 bp human Fth1 fragment was cloned into plasmid pL5652 and integrated at the yeast TDH3 promoter in strain BY4741 to produce strain P3190. To characterize the human heavy chain ferritin (Fth1) expressed in the P3190 yeast cell line, it was grown in high iron-containing media, and after cell lysis, the Fth1 protein was purified from soluble lysates to >90%. Based on ferrozine assays, it contained significant amounts of iron, estimated at ~110 Fe atoms per subunit. Size Exclusion Chromatography column-purified fractions analyzed by electron microscopy revealed 4–8 nm dense particles with the properties expected for Fth1 as a 24-mer. Collectively, the analysis indicated the Fth1 in this yeast formed native oligomers containing non-mineralized iron rich cores.

7. Can bile salts with alpha OH groups and with a carboxylate group from the side chain complex the Fe3+ ion so that the Fe3+ ion would participate in the enterohepatic circulation.

Studies in rats showed that a bile salt (e.g., taurocholate) can enhance iron absorption in the upper small intestine (Sanyal et al., 1991, 1992, 1994) The alpha OH and carboxyl (COOH) groups – and their adjacent location - are apparently important for high affinity Fe2+ (and Ca2+) binding. However, this effect was observed for Fe2+ and not Fe3+. Furthermore, some aspects of enhanced iron absorption with bile salts may be species-specific and is not seen in humans. We also need to be cautious about applying this information to the monkey model of infantile anemia because our previous metabolomic analysis of serum bile indicated that circulating levels of taurocholate concentrations were 70% lower in iron-deficient infants.  In addition, it is generally believed that enterophepatic circulation plays a more important role in iron excess (i.e., hemochromatosis) than in iron sufficiency and deficiency. Thus, it will require additional research to further resolve the importance of this pathway in the primate model.

  1. Sanyal AJ, Shiffmann ML, Hirsch JI, Moore EW. Premicellar taurocholate enhances ferrous iron uptake from all regions of rat small intestine. 1991, 101(2), 382-9.
  2. Sanyal AJ, Hirsch JI, Moore EW. High-affinity binding is essential for enhancement of intestinal Fe2+ and Ca2+ uptake by bilesalts. 1992, 102(6), 1997-2005.
  3. Sanyal AJ, Hirsch JI, Moore EW. Evidence that bilesalts are important for iron absorption. J. Physiol., 1994, 266(2 Pt 1), G318-23.
  4. Sandri BJ, Lubach GR, Lock EF, Georgieff MK, Kling PJ, Coe CL, Rao RB. Early-Life IronDeficiency and Its Natural Resolution Are Associated with Altered Serum Metabolomic Profiles in Infant Rhesus Monkeys. J Nutr. 2020, 150(4), 685-693. doi: 10.1093/jn/nxz274.PMID: 31722400 

8. The figures from the manuscript would be attractive, but especially the C and D parts are not visible, the markings and text are unreadable (Fig 1-3).

As mentioned in the Overall Response to the Reviewers, the figures now have only 3 panels rather than 4 panels, which allowed us to expand Panel C to extend the full height of the image.  The 4th panel, which showed the same data in cladogram format has been relocated to Supplementary Materials, where it could be printed as a separate illustration on a full page.

Reviewer 3 Report

Comments and Suggestions for Authors

Coe et al.

Iron deficient infants at 6mo of age.

16S seq (v?) Lefse and richeness. Then given iron. Alpha increased. Were they all given iron?

Importance of iron etc.

21 young rhesus monkeys. UW Madison. Muscle injections and b 12. Yeast with ferritin. V4.

Figure 1 a bit blurry, make sure quality is high for pub. I’d recommend not using the output from emperor as a figure, the beta diversity should be examined using R.

I think the paper would benefit from having the Picrust analyses IN the main text. Can these be incorporated? Can S2-4 be merged?

I do agree with the limitation addressed in lns 447 etc but I appreciate that the authors mentioned them and I don’t feel that they nullify the results of the study in any capacity. It is very well done.

Overall, I find this study to be well designed and manuscript to be thoughtfully written. The authors take the time to address the many impacts of iron deficiency on primates in a controlled setting. The discussion is detailed and accurately encompasses the results documented therein. I am happy to support this manuscript but with just a few comments I have above, and look forward to seeing more work from this group in the future.

Author Response

Reviewer 3.

We appreciate that Reviewer 3 felt the study was well designed and thoughtfully written with sufficient detail.  We have responded to the reviewer’s suggestions and concerns in the revision. As mentioned above, the figures have been reformatted and presented with more clarity.

  1. Were they all given iron?

As mentioned in the response to Reviewer 1, we have now clarified more clearly that the iron treatments were initiated after the initial screening of the 21 monkeys and the determination that a subset of the infants were iron-deficient anemic.  The approval of our study by the institutional Animal Care and Use Committee (ACUC) was based on a compliance agreement that we would treat any monkey found to be anemic during the initial screening until its hematology had returned into the normal range. Thus, only the anemic monkeys were administered iron or given oral iron supplements. This aspect of the experimental design is now conveyed more clearly in the Abstract.  In addition to allowing us to address the scientific aims, we were being mindful of animal welfare considerations and the importance of remediating the anemia after it was detected. The effects of the two iron treatments on the gut microbiome were evaluated by comparing the changes to the baseline anemic state as well as by comparison between the two treatments.

  1. I think the paper would benefit from having the Picrust analyses in the main text. Can these be incorporated? Can S2-4 be merged?

In response to this request, we have relocated the PICRUSt analysis of inferred functional pathways that might be differentially affected in anemic infants as compared to iron-sufficient to the Results section.  It is now Figure 4.  However, given the concerns about legibility and font size, we kept the PICRUSt analyses examining the effects of iron treatments in Supplementary Materials, where they could be presented more legibly on a full page.  If all were presented together as 3 panels in one figure, it would recreate the concerns about legibility of the terms and descriptors.  In addition, the comparison of anemic infants to iron-sufficient infants provided the largest N.

We are very appreciative of the reviewer’s hope to see this research extended in the future.